# The Pattern and Progression of “Usual” Interstitial Pneumonia with Autoimmune Features: Comparison with Patients with Classic Interstitial Pneumonia with Autoimmune Features and Idiopathic Pulmonary Fibrosis

**DOI:** 10.3390/jcm13020369

**Published:** 2024-01-10

**Authors:** Alessandro Libra, Michele Colaci, Lucia Spicuzza, Giuliana Luca, Sefora Fischetti, Giorgio Pashalidis, Chiara Alfia Ferrara, Giuseppe Ielo, Domenico Sambataro, Giuliana La Rosa, Federica Libra, Stefano Palmucci, Carlo Vancheri, Gianluca Sambataro

**Affiliations:** 1Department of Clinical and Experimental Medicine, Regional Referral Center for Rare Lung Disease, Policlinico “G. Rodolico-San Marco”, University of Catania, 95123 Catania, Italy; alessandrolibra@outlook.it (A.L.); lucia.spicuzza@unict.it (L.S.); giulianaluca97@gmail.com (G.L.); sefora.fischetti@gmail.com (S.F.); giorgiopashalidis@gmail.com (G.P.); chiara.ferrara@virgilio.it (C.A.F.); peppeiellow@gmail.com (G.I.); vancheri@unict.it (C.V.); 2Internal Medicine Unit, Department of Clinical and Experimental Medicine, Division of Rheumatology, Cannizzaro Hospital, University of Catania, 95123 Catania, Italy; michele.colaci@unict.it; 3Artroreuma s.r.l., Rheumatology Outpatient Clinic, 95030 Mascalucia (CT), Italy; d.sambataro@hotmail.it; 4Department of Medical Surgical Sciences and Advanced Technologies “GF Ingrassia”, University Hospital Policlinico “G. Rodolico-San Marco”, 95123 Catania, Italy; giulianalarosa4@gmail.com (G.L.R.); federica.libra@hotmail.it (F.L.); 5Department of Medical Surgical Sciences and Advanced Technologies “GF Ingrassia”, University Hospital Policlinico “G. Rodolico-San Marco”, Unità Operativa Semplice Dipartimentale di Imaging Polmonare e Tecniche Radiologiche Avanzate (UOSD IPTRA), 95123 Catania, Italy; spalmucci@unict.it

**Keywords:** interstitial pneumonia with autoimmune features, idiopathic pulmonary fibrosis, usual interstitial pneumonia, connective tissue disease, diagnosis

## Abstract

Background: We proposed the term “UIPAF” to define patients with Usual Interstitial Pneumonia (UIP) associated with only one domain of the classification called “Interstitial Pneumonia with Autoimmune Features” (IPAF). The objective of this study was to evaluate the clinical presentation and prognosis of UIPAF patients, compared with two cohorts, composed of IPAF and idiopathic pulmonary fibrosis (IPF) patients, respectively. Methods: The patients were enrolled as IPAF, UIPAF, or IPF based on clinical, serological, and radiological data and evaluated by a multidisciplinary team. Results: We enrolled 110 patients with IPF, 69 UIPAF, and 123 IPAF subjects. UIPAF patients were similar to IPAF regarding autoimmune features, except for the prevalence of Rheumatoid Factor in UIPAF and anti-SSA in IPAF. A similar proportion of the two cohorts progressed toward a specific autoimmune disease (SAD), with differences in the kind of SAD developed. The real-life management and prognosis of UIPAF patients proved to be almost identical to IPF. Conclusions: UIPAF shared with IPAF similar autoimmune features, suggesting the opportunity to be considered IPAF, excluding the morphological domain by the classification. However, the real-life management and prognosis of UIPAF are similar to IPF. These data suggest a possible modification in the therapeutic management of UIPAF.

## 1. Introduction

In 2015, the European Respiratory Society (ERS) and American Thoracic Society (ATS) “Task Force on Undifferentiated Forms of Connective Tissue Disease-associated Interstitial Lung Disease” proposed classification criteria called Interstitial Pneumonia with Autoimmune Features (IPAF). The classification aims to include patients with interstitial lung disease (ILD) who manifest autoimmune features but do not fulfill the diagnostic criteria for a definite connective tissue disease (CTD) [1,2,3]. These classification criteria were based on a combination of features from three domains: a clinical domain consisting of extra-thoracic features; a serologic domain with specific autoantibodies; and a morphologic domain with imaging patterns, histopathological findings, or multi-compartment involvement.

Since the publication of the classification criteria, several studies have tried to evaluate the prognosis of IPAF patients with conflicting results, probably due to the retrospective nature of the data. IPAF seems to have a better prognosis compared to Idiopathic Pulmonary Fibrosis (IPF) and, in general, other idiopathic ILDs, but worse than CTD-ILD [4,5,6]. Huapaya JA et al. identified a stable disease in IPAF; however, the study retrospectively involved a limited number of patients, with a rate of progression toward specific autoimmune diseases (SADs) of 60% [7]. The proportion is about 3 times higher than that reported in previous studies, suggesting the presence of pre-existing (albeit occult) conditions [4,8,9,10]. The prognosis of IPAF patients seems to be closely associated with the presence of a Usual Interstitial Pneumonia (UIP) pattern on chest high-resolution computed tomography (HRCT). UIP-IPAF patients proved to have a prognosis similar to idiopathic pulmonary fibrosis (IPF), whereas non-UIP-IPAF were similar to CTD-ILD [11]. It should be considered that IPAF patients with a UIP definite pattern are less common due to the definitions used in the criteria. Despite a UIP definite pattern being prevalent in ILD associated with rheumatoid arthritis (RA) and vasculitides and the second most common pattern in CTDs [12], the authors deemed this pattern not to be sufficiently associated with autoimmune ILD. For this reason, IPAF criteria allow the enrolment of UIP patients in the presence of at least one item from both the clinical and serological domains [3]. However, the majority of IPAF patients are enrolled with a minimum of two criteria, in which the morphological domain is almost always present [10]. To evaluate the diagnostic weight of the morphological domain, we proposed a classification called “Usual Interstitial Pneumonia with Autoimmune Features” (UIPAF) [8]. We defined UIPAF as those patients with a UIP definite pattern associated with items belonging to only one IPAF domain (clinical or serological). UIPAF patients proved to have a rate of progression toward SADs that was significantly higher than IPF and similar to IPAF. Thus, we suggested removing the morphological domain from the IPAF criteria [8,13]. However, to date, limited data have been reported on the prognosis of UIPAF patients. In a previous study involving 11 UIPAF patients, these patients showed a higher rate of progression of lung damage compared with two different cohorts, including IPAF and CTD-ILD patients [14]. 

The aim of this study is to describe the prognosis of a cohort of UIPAF patients compared with two different cohorts composed of patients with IPF and classic IPAF.

## 2. Materials and Methods

The study was conducted prospectively in the Regional Referral Center for Rare Lung Disease, University of Catania, and approved by our local ethical committee (n.0024182TMP/10-2015) from our registry of IPAF patients actively enrolled since January 2017. The study complies with the Declaration of Helsinki, and written informed consent was obtained from all the patients enrolled.

### 2.1. Patients

All ILD patients referred to our center were screened for potential inclusion in the IPAF/UIPAF registry. To define patients with IPAF, we exploited the tight collaboration that exists between our rheumatologists and pulmonologists working together on the same staff. Pulmonologists managed the respiratory clinic, collecting information regarding respiratory symptoms, exposure, and smoking habits, quantified using pack/years. Pulmonologists also evaluated patients in order to classify patients for possible known causes of ILD. Rheumatologists were asked to recognize the presence of the items included in the clinical domain of IPAF criteria, as well as to exclude the presence of an underlying SAD. All the diagnoses of IPAF were collegially discussed by the multidisciplinary team. It should be noted that patients classified as UIPAF, according to our definition, were managed as IPF until the possible recognition of an SAD. Patients satisfying criteria for SAD were excluded [15,16,17,18,19,20,21,22,23]. The presence of antisynthetase syndrome (ASSD) was defined by the criteria proposed by Connors GR et al. [24]. IPF patients were enrolled consecutively from January 2019 to June 2022. The diagnosis of IPF was established according to the latest guidelines [25]. 

IPAF criteria were collected, when present, in each patient. The presence of polyarticular morning joint stiffness of at least one hour was objectively considered present if associated with increased values of erythrosedimentation rate and/or C Reactive protein (CRP) [26].

Patients enrolled as IPAF, UIPAF, or IPF were scheduled for trimestral or semestral follow-up, according to clinical needs. From the prognostic point of view, data on mortality were collected from our database or communicated by telephone by the caregiver. The progressive-fibrosing phenotype was defined according to ATS criteria [25]. Treatment with antifibrotics or immunosuppressants was defined by at least 6 months of treatment at the therapeutic dosage for each drug. Treatment with prednisone was considered an immunosuppressant treatment at the minimum dosage of 10 mg/die. 

### 2.2. Serological Assessment

All the patients were first evaluated with a first-line assessment including complete blood count, ESR, CRP, transaminases, creatinine, lactic dehydrogenase, creatine phosphokinase, myoglobin, aldolase, complement fractions C3 and C4, serum protein electrophoresis, Rheumatoid Factor (RF), anti-neutrophilic cytoplasm antibodies, anti-citrullinated protein antibodies (ACPA), antinuclear antibodies (ANA) in indirect immunofluorescence with a description of the pattern, and an anti-extractable nuclear antigen panel. The latter panel included anti-Jo1, anti-La, anti-Ro60kD and anti-Ro52kD, anti-Sm, anti-RNP, and anti-Scl70. Second-line serological assessment was performed on patients with a suspicion of scleroderma-related disorders if seronegative for anticentromeric antibodies or anti-Scl70, as well as in patients with a suspicion of idiopathic inflammatory myopathy (IIM). The suspicion of systemic sclerosis (SSc) was based on the presence of Raynaud’s Phenomenon, puffy fingers, or positivity to Nailfold videocapillaroscopy [27]. The suspicion of IIM was defined by proximal, symmetric asthenia, presence of a combined ILD radiological pattern of nonspecific and organizing pneumonia (NSIP + OP), unexplained fever, or increased level of muscle enzymes of at least 1.5× the upper limit [26]. The second panel was performed using immunoblotting and included anti-mi2, anti-MDA5, anti-NXP2, antiSAE1, anti-SRP, anti-Tif1γ, anti pm/scl, and antisynthetase antibodies. The general assessment was periodically repeated on patients based on their clinical needs, while the autoimmune assessment was repeated annually. All the autoantibodies included in the serological domain of IPAF were included in the assessment; however, patients with anti-synthetase antibodies and ILD were classified as ASSD according to the criteria used [24], also considering that the inclusion of these autoantibodies in the IPAF criteria is questioned [28]. Isolated positivity for anti-La, although considered in IPAF criteria, was not considered sufficient, as the lack of association between anti-La alone and autoimmunity has already been established [29].

### 2.3. Instrumental Evaluation

All patients were assessed with an HRCT, interpreted by an expert radiologist, and classified according to the latest guidelines [25]. The HRCT was repeated every 12 months or when clinically needed.

Nailfold videocapillaroscopy (NVC) was performed by expert rheumatologists with VideoCap 10.00.14, DsMedica, Milan, Italy), on all patients with Raynaud’s Phenomenon (RP) or a suspicion of IIM and scleroderma. Interpretation was undertaken according to the latest guidelines [30]. Rheumatologists also performed tests to evaluate the function of exocrine glands (e.g., Schirmer’s test) in the suspicion of primary Sjogren’s Syndrome (pSS).

Pulmonary function tests and six-minute walking tests were conducted by pulmonologists according to the latest guidelines [31,32,33,34]. Values of forced vital capacity (FVC) and diffusion lung capacity for carbon monoxide (DLCO) were reported in proportion to the predicted.

Other instrumental exams such as biopsy of the lung/salivary glands/muscle or magnetic resonance were performed when deemed useful for the appropriate management of the patients.

### 2.4. Inclusion and Exclusion Criteria

Patients were enrolled in the presence of a written informed consent to be included in the registry and the study. For the aim of this study, we enrolled patients with a minimum follow-up of 12 months. The minimum assessment was the rheumatologic evaluation within 3 months of the first and last visit, the presence of an HRCT at the baseline, and a complete first-line serological assessment. Patients without these inclusion criteria were excluded from the study. A flowchart is reported in Figure 1.

### 2.5. Statistical Evaluation

Statistical analysis was performed with IBS SPSS statistics for Windows v.20.0 (Armonk, NY, USA). We used a D’Agostino and Pearson test to evaluate the distribution of the data. Continuous variables were evaluated using the Kruskal–Wallis test for independent data, whereas paired data were evaluated with the Friedman test for the analysis of the two-way rank variance. Dichotomous variables were evaluated using the X^2^ test. Descriptive variables were reported in mean (±standard deviation, SD), whereas dichotomous variables were reported in proportion. Differences were considered statistically significant with a *p* ≤ 0.05. The mortality was evaluated by the Kaplan–Meier survival curve and was estimated to compare survival between groups.

## 3. Results

After the application of the inclusion and exclusion criteria, we enrolled 110 patients with IPF, 69 UIPAF, and 123 IPAF subjects. The UIPAF cohort had a mean age of 68.5 ± 7 years, similar to IPAF (66.1 ± 10.2 years, *p* = 0.46) and IPF (70.2 ± 7.5 years, *p* = 0.41). IPF patients were significantly older than IPAF (*p* = 0.002). The highest male proportion was reported in the IPF cohort (76.1%), similar to UIPAF (63.8%, *p* = 0.09), in both cases higher than IPAF (40.7% vs. UIPAF *p* = 0.003 vs. IPF *p* ≤ 0.0001). The IPF and UIPAF groups were also similar in the proportion of current or former smokers (71 and 71.6%, respectively), greater than that reported for IPAF (53.7%, *p* = 0.02 vs. both). However, the quantification of pack/years of smoking habit was similar: IPF 36.6 ± 39.2, UIPAF 45.5 ± 34.8 vs. IPAF 34 ± 34.4, *p* = 0.06.

The prevalence of IPAF criteria in the UIPAF and IPAF cohort is reported in Figure 2.

The UIPAF and IPAF cohorts showed a similar proportion of ANA positivity (27.5% and 35.8%, respectively, *p* = 0.26). Also, the distribution of the patterns was similar: In the two cohorts, the proportion of a homogeneous pattern was 5.8% and 6.5% (*p* = 1), the speckled pattern was noted in 11.6% of UIPAF patients and 20.3% IPAF subjects (*p* = 0.16), and the nucleolar pattern in 4.3% and 13%, respectively (*p* = 0.08). No patients with UIPAF reported the centromeric pattern, whereas in IPAF, it was reported in 1.6% (*p* = 0.08). Finally, a cytoplasmic pattern was found in 5.8% of UIPAF patients and 9.8% of those with IPAF (*p* = 0.43). Clearly, all the patients with UIPAF reported a Usual Interstitial Pneumonia (UIP) definite pattern, while it was present in only 6.5% of IPAF (*p* ≤ 0.0001). IPAF patients also showed a UIP probable pattern in 12.2%, NSIP in 65.9%, OP in 8.9%, and other patterns in 12.2%.

The rate of progression toward an autoimmune disease was similar in the two groups (22.4% in UIPAF, 23.6% in IPAF *p* = 0.84). IPAF progressed toward Primary Sjögren’s Syndrome (pSS) in 11 cases and IIM in 14. Other progressions were toward microscopic polyangiitis (MPA), RA, SSc, and a case of overlap condition SSc+ systemic lupus erythematosus (SLE). UIPAF patients progressed toward RA and MPA in 6 and 5 cases, respectively. Two other patients developed SSc, whereas other conditions were pSS and a SLE (one for each condition). Data are reported in Figure 3.

From a functional point of view, at the baseline, the three groups showed similar values of FVC, but IPAF showed significantly higher values of DLCO. During the 24-month follow-up, values of FVC were stable in IPAF (FVC *p* = 0.23, DLCO *p* = 0.06). A significant functional impairment was noted in UIPAF for both FVC (*p* = 0.02) and DLCO (*p* = 0.048). The IPF cohort reported a significant impairment of FVC (*p* ≤ 0.0001) but not of DLCO (*p* = 0.08). Figure 4 reports the mean values of functional parameters in the two cohorts.

Patients with IPAF were treated with immunosuppressants in 78% and antifibrotics in 9.8% of cases. UIPAF and IPF patients were instead treated with immunosuppressants in 24.6% and 11.9%, and antifibrotics in 88.4% and 97.2%, respectively (*p* ≤ 0.0001 vs. IPAF). Details on treatment are reported in Table 1.

The need for oxygen support was present from the first visit in 52.2% of UIPAF patients, 36.7% of IPF subjects, and 20.5% of IPAF patients. After one year of follow-up, the proportions increased to 64.7%, 54.4%, and 26.8%, respectively. At the end of follow-up, the proportion was 70.6% for IPF and 69.6% for UIPAF, whereas IPAF patients required oxygen in 46.3%. The mean time to the development of oxygen need was 10.3 ± 12.6 months in IPF, 15.5 ± 14.9 in UIPAF, and 16.2 ± 19.2 in IPF (*p* = 0.06). No differences in the necessity of oxygen were noted between UIPAF and IPF, whereas the proportion was significantly higher in both groups than in IPAF (*p* = 0.0001 for all).

ATS criteria to define the progressive-fibrosing phenotype were reached by 69.6% of UIPAF, 49.5% of IPF, and 37.4% of IPAF patients (*p* ≤ 0.0001). The mean time to reach the ATS criteria was 13.9 ± 5.3 months for IPAF, 16.8 ± 6.4 in UIPAF, and 16.9 ± 6.1 in IPF (*p* = 0.02) (Figure 5).

Hospitalizations were 18.7% in IPAF, 14.5% in UIPAF, and 11.9 in IPF (*p* = 0.35); acute exacerbations were 9.8%, 4.3%, and 9.2%, respectively (*p* = 0.39). We registered 27% of death in the IPF cohort, similar to UIPAF (30%) but higher than IPAF (8%, *p* ≤ 0.0001). The mean time from the diagnosis to death for respiratory causes (progression of fibrosis or acute exacerbations) was 33.4 ± 27.8 months in IPAF, 42.8 ± 24.2 in UIPAF, and 32.5 ± 17.9 in IPF (*p* = 0.19). After adjusting for age and gender, we analyzed survival via the Kaplan–Meier curve, based on treatment and functional impairment. The presence of a functional impairment, defined by the presence of FVC < 50% or DLCO < 36% of the predicted, was associated with a worse prognosis in IPF and UIPAF but not in IPAF, which showed a better prognosis (Figure 6).

Finally, we evaluated the association between IPAF criteria and the PF phenotype. We noted only a possible protective effect of mechanic’s hands in UIPAF patients (*p* = 0.02, X^2^ = 7.2). No IPAF items were associated with death in IPAF or UIPAF cohorts.

## 4. Discussion

The diagnosis of SADs in patients with ILD can be very challenging for several reasons. First of all, some conditions, such as IIM, could have ILD without any other sign of the disease during the follow-up [35]. In other conditions such as pSS, ILD can be the first manifestation, even associated with a UIP pattern, mild symptoms, and, less commonly, seropositivity: these patients are very difficult to distinguish from an IPF [36]. Even when the other clinical features are present, they are milder than those reported in the same condition in other subsets [34]. However, the recognition of an SAD underlying ILD is crucial for therapeutic objectives, to prevent other clinical manifestations, and to exploit the possible role of immunosuppressants to slow, stop, or even improve lung damage.

Since their proposal, IPAF criteria have played a pivotal role in the recognition of autoimmune-mediated ILD, serving as “red flags” of autoimmune disease in all the respiratory units dedicated to ILD patients. Several retrospective studies reported a large prevalence of the NSIP pattern in IPAF patients [37,38,39,40]; however, these data are due to the necessity of having both the clinical and the serological domains associated with a UIP pattern to be included as IPAF. We believe that the recognition of an autoimmune pathogenic pathway underlying disease could also be useful in UIP patients, and therefore, we proposed removing the morphological domain from IPAF criteria. To demonstrate this, we prospectively collected UIP patients with only one IPAF domain, calling this group UIPAF. In this study, we evaluated the prognosis of UIPAF patients with two different groups, composed of patients with IPAF and IPF.

As expected, IPF was the oldest cohort, whereas the UIPAF cohort reported intermediate values between IPAF and IPF. For other general parameters, such as the proportion of smokers and the proportion of male patients, UIPAF and IPF proved to be similar, with a significant difference with IPAF. These results, as expected, were increased age, male gender, and smoking habits established risk factors for the UIP pattern [41].

Consistent with the previous literature, UIPAF patients are very similar to the classic IPAF regarding autoimmune features. Beyond the HRCT pattern of ILD, the only difference is the prevalence of RF in UIPAF and anti-SSA in IPAF. A very similar proportion of the two cohorts progressed toward an SAD, with differences only in the kind of SAD. In the IPAF group, the patients mainly progressed toward IIM and pSS, whereas in UIPAF, toward MPA and RA. This difference can be explained by the radiological pattern of the underlying ILD. UIP is the most common pattern in MPA and RA, whereas the NSIP+/-OP pattern is very common in IIM and pSS [26]. Also, serologically, MPA and RA commonly show positivity for RF, whereas anti-SSA (60kD and 52kD) are associated with the diagnosis of IIM and pSS [15,16,17,18,19,20,21,22,26].

While the clinical presentation of UIPAF was very similar to IPAF, the prognosis proved to be almost identical to IPF. Actually, IPF and UIPAF are similar to each other in final prognosis and baseline DLCO, with a statistical difference with IPAF. UIP pattern is associated with a poor prognosis in both idiopathic and secondary ILD [26,42]. However, management of the conditions is different. While immunosuppression was associated with increased mortality in IPF, immunosuppressive treatment was found to improve pulmonary functions in UIP associated with RA and CTD-ILD [43,44,45]. Similar results were also obtained in IPAF patients: treatment with Micophenolate Mofetil and Prednisone was associated with non-progression in IPAF patients, even controlling for the presence of a radiological pattern of UIP [46,47]. Recently, Yamano Y et al. reported an interesting study on histologically proven UIP-IPAF patients, comparing the treatment of these patients [48]. In the study, antifibrotic treatment stabilized the disease, while immunosuppressant treatment improved pulmonary function. The response to immunosuppressant treatment was associated with a significant presence of inflammatory cells in the context of a histological UIP pattern. Unfortunately, it is impossible to propose a biopsy on all ILD patients; however, IPAF criteria could be very useful if they improve their ability to identify patients in which an immunosuppressive treatment could be useful even with a UIP pattern.

Treatment is outside the scope of this study, and the classification as UIPAF did not influence the management of these patients. UIP patients with insufficient autoimmune items to be classified as SADs were clinically diagnosed as IPF and treated accordingly. Therefore, the vast majority of UIPAF patients were mainly treated with antifibrotics, using immunosuppressants on the development of an SAD. The possible role of immunosuppressant treatment in UIPAF patients should be assessed in prospective studies. The IPAF and UIPAF cohorts were actually very similar in their autoimmune presentation, and the values of FVC were also similar at the baseline and after the 1-year follow-up. We cannot exclude the possibility that at least a subset of UIPAF patients could benefit from immunosuppression, as has already been proven in IPAF [48]. Histological exams could not only provide useful information for the treatment of these patients [48] but also increase knowledge on the clinical presentation of autoimmune disease presenting with ILD as the first sign. On that, it is appropriate to remember that in 2005, an important clinical trial suggested the possible role of high doses of acetylcysteine, prednisone, and azathioprine in the treatment of IPF [49]. Unfortunately, another clinical trial directly aiming to evaluate this triple therapy in IPF, despite having enrolled patients with similar clinical features to those enrolled in the previous study, identified a detrimental effect [42]. The second study started enrolment in 2010, the same year as the publication of the new RA classification criteria. These criteria are able to classify more patients with RA and even at an earlier phase, including ESR, CRP, and above all, ACPA [50]. Considering that UIP-ILD is a possible clinical onset of RA (but also CTDs and vasculitides) [26], and the proportion of patients that progressed toward RA in the UIPAF cohort presented in this study, we cannot exclude the possibility that the study performed by Demedts M et al. included some, possibly occult, forms of RA [49]. This proportion might explain the improvement seen in IPF patients in the first trial and, in this case, could also support this treatment in UIPAF patients.

The study has some limitations: first of all, the classification of patients is based on radiological data, and histological assessment was performed only on a minority of patients. The number of UIPAF patients treated with immunosuppressants is too low for any possible speculation. However, the study also has some merits. To the best of our knowledge, to date, this is the largest study aimed at evaluating the prognosis of UIPAF patients. It was conducted in a center with a well-established collaboration between pulmonologists and rheumatologists with great expertise in ILD; therefore, the risk of including occult, definite forms of SADs as IPAF is low. Finally, the data were collected from a prospective, longitudinal registry, actively enrolling patients from 2017.

## 5. Conclusions

Our study showed that in both UIPAF and IPAF patients, there is a risk of progression toward SADs, but with a difference in terms of the pathology into which it can evolve, probably in part suggested by different radiological patterns. The radiological pattern plays a pivotal prognostic role, as the prognosis of UIPAF patients is similar to those with IPF. We cannot exclude the possibility that adding immunosuppressive treatment to the standard of care could improve, to some extent, the prognosis of UIPAF patients. New revisions of IPAF criteria should take into consideration the possibility of removing the morphological domain, thus classifying UIPAF and IPAF together in the same definition.

## Figures and Tables

**Figure 1 jcm-13-00369-f001:**
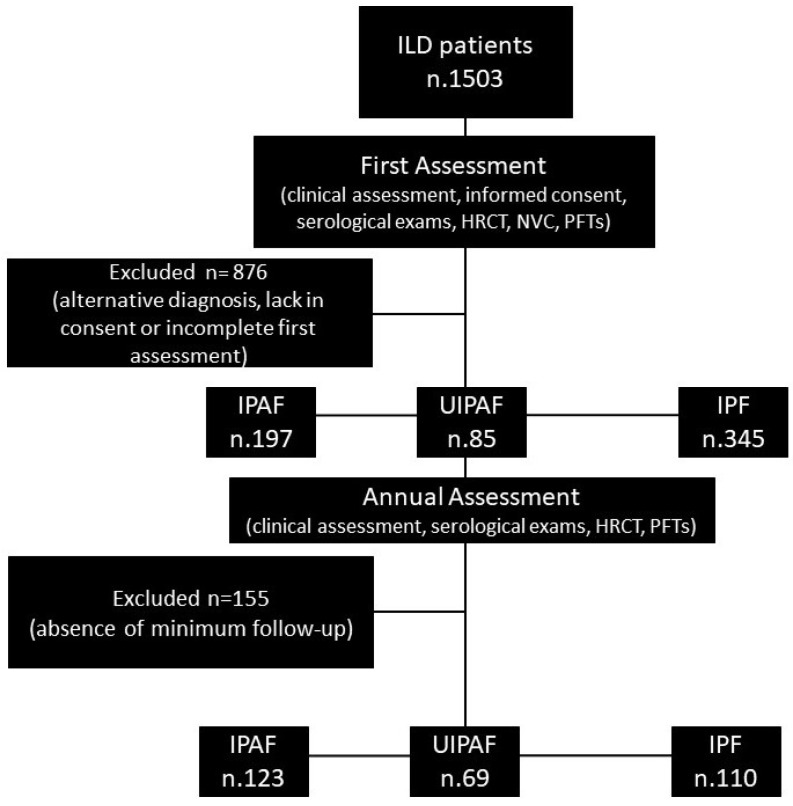
Flowchart of patient selection. Legend: HRCT: high-resolution computed tomography; IPAF: Interstitial Pneumonia with Autoimmune Features; IPF: idiopathic pulmonary fibrosis; NVC: nailfold videocapillaroscopy; PFTs: pulmonary function tests; UIPAF: Usual IPAF.

**Figure 2 jcm-13-00369-f002:**
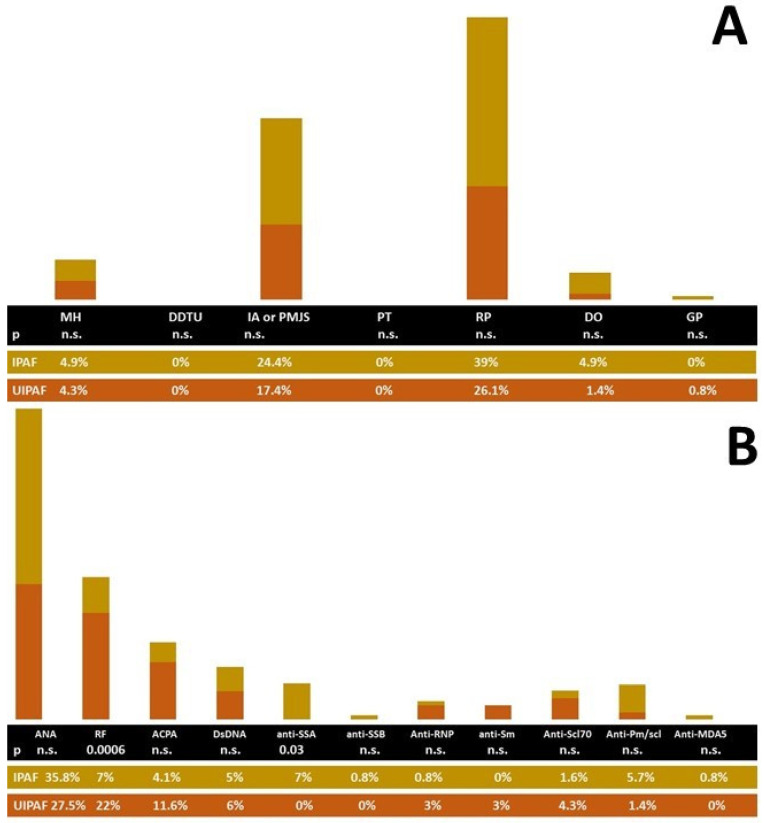
Distribution of the clinical (**A**) and serological (**B**) IPAF items in the cohort with IPAF and UIPAF. Legend: ANA: antinuclear antibodies; ACPA: anti cyclic citrullinated antibodies; DDTP: distal digital tip ulcerations; DO: digital oedema; GP: Gottron’s Papules; IA: inflammatory arthritis; NSIP: nonspecific interstitial pneumonia; OP: organizing pneumonia; PMJS: polyarticular morning joint stiffness of at least 1 h; PT: palmar telangiectasia; RF: Rheumatoid Factor.

**Figure 3 jcm-13-00369-f003:**
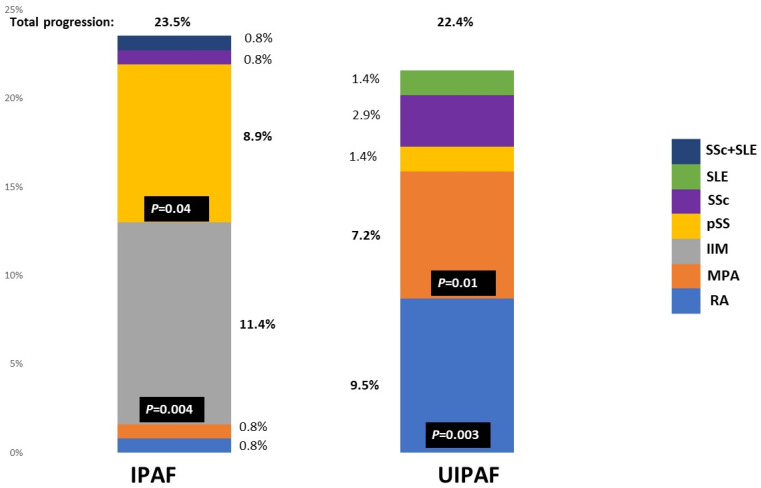
Progression toward specific autoimmune disease of the IPAF and UIPAF cohort. Legend: IIM: idiopathic Inflammatory myopathies; IPAF: Interstitial Pneumonia with Autoimmune Features; MPA: micro polyangiitis; pSS: primary Sjögren’s Syndrome; UIPAF: Usual Interstitial Pneumonia with Autoimmune Features; RA: rheumatoid arthritis; SLE: systemic lupus erythematosus; SSc: systemic sclerosis.

**Figure 4 jcm-13-00369-f004:**
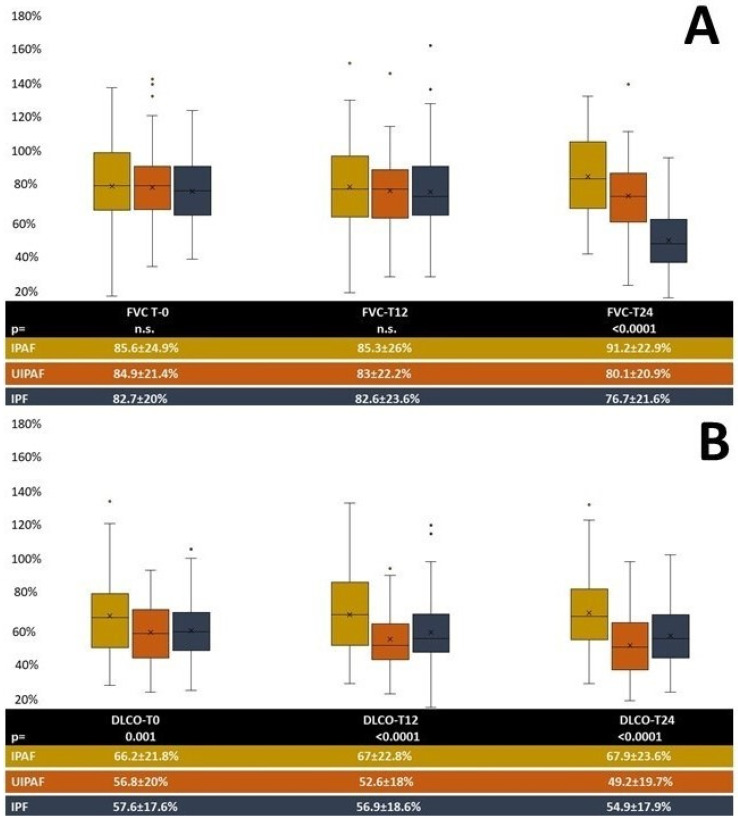
Mean values of FVC (**A**) and DLCO (**B**) during follow-up in the two cohorts. Legend: DLCO: diffusion lung capacity for carbon monoxide; FVC: forced vital capacity, T0: baseline; T-12: first follow-up at 1 year; T-24: second follow-up at 2 years.

**Figure 5 jcm-13-00369-f005:**
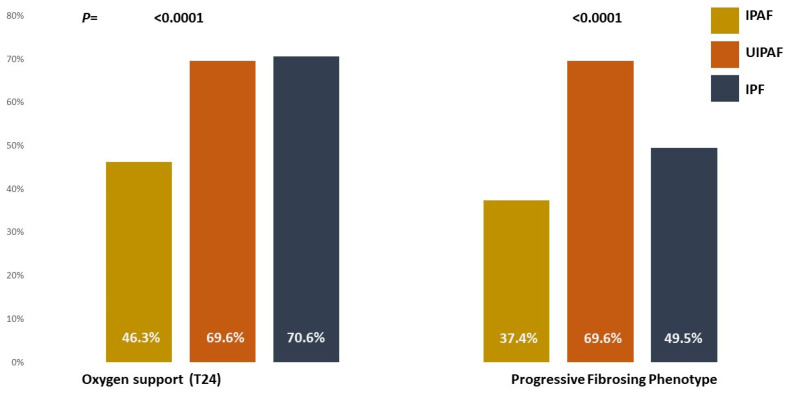
Outcome of the three cohorts at the end of the follow-up. Legend: progressive-fibrosing phenotype: proportion of patients satisfying the ATS criteria at the end of the follow-up; T24: last check at the end of follow-up, 2 years.

**Figure 6 jcm-13-00369-f006:**
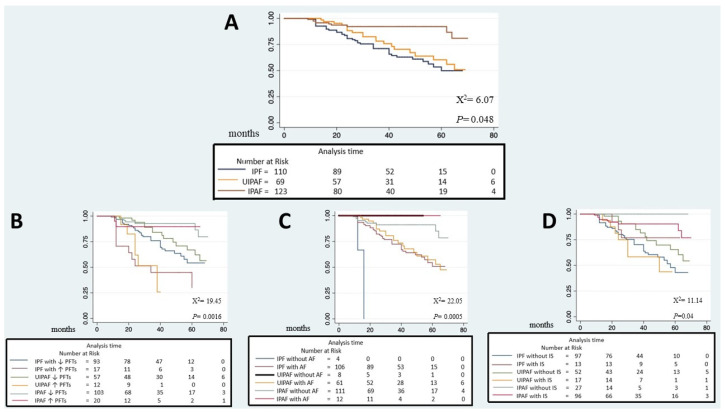
Kaplan–Meier survival curve of the three cohorts. (**A**): crude mortality after adjusting for age and gender; (**B**): mortality of the three cohorts based on the functional impairment; (**C**): mortality of the three cohorts stratified according to the treatment with antifibrotics; (**D**): mortality of the three cohorts based on the treatment with immunosuppressants; Legend: AF: antifibrotic; IS: immunosuppression; PFTs: pulmonary function tests ↓: lower; ↑: higher.

**Table 1 jcm-13-00369-t001:** Drugs used in the treatment of the cohorts studied.

Drugs	IPAF n = 123	UIPAF n = 69	IPF n = 110
Prednisone ≥ 10 mg/die	71.5%	23.2%	11.8%
Azathioprine	24.4%	0%	0%
Micophenolate mofetil	22.8%	2.8%	0%
Cyclosporine A	1.6%	1.4%	0%
Methotrexate	6.5%	1.4%	0%
Rituximab	0.8%	0%	0%
Combined immunosuppression	39.8%	4.3%	0%
Nintedanib	6.5%	43.5%	45.5%
Pirfenidone	3.2%	55.1%	54.5%

Legend: IPAF: Interstitial Pneumonia with Autoimmune Features; IPF: idiopathic pulmonary fibrosis; UIPAF: Usual Interstitial Pneumonia with Autoimmune Features.

## Data Availability

The data presented in this study are available upon request from the corresponding author. The data are not publicly available due to ethical reasons.

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
