# Peer review of "The Pattern and Progression of “Usual” Interstitial Pneumonia with Autoimmune Features: Comparison with Patients with Classic Interstitial Pneumonia with Autoimmune Features and Idiopathic Pulmonary Fibrosis"

_jcm, 2024, doi:10.3390/jcm13020369_

Round 1

Reviewer 1 Report

Comments and Suggestions for Authors

In this study, the authors analyse a large cohort of patients with idiopathic pulmonary fibrosis (IPF), interstitial pneumonia with autoimmune features (IPF) and a peculiar IPAF subtype, namely UIPAF (IPAF with a UIP pattern on HRCT).

As compared to IPF, UIPAF and IPF displayed similar rate of antibodies and a similar rate of evolution towards a specific autoimmune disease (20-25%).

On the other hand, IPF and UIPAF were characterized by the use of antifibrotics and a propensity to progress according to the recent ATS-ERS criteria.

General comment:

Although the findings are incremental (UIP pattern is associated with worse outcome), this study explores an interesting and yet poorly explored area (UIPAF). I think the authors should address the following concerns:

Main concerns:

- Study population: considering the 2015 position paper, UIPAF should be considered as a subset of IPAF rather than a specific population. This point should be highlighted by the authors. Furthermore, it is striking that 22% of UIPAF have a positive rheumatoid factor and 11,7% positive ACPA, altogether pointing towards early rheumatoid arthritis. This point should be discussed.

- Study design: in its present form, it is not very clear whether this is an observational propspective study of a retrospective study. The authors should clarify this and provide a study FlowCHART.

- A strong effort should be done on the clarity of figures: for example, Figure 4 is also noted as fiigure 3, some abbreviations are not defined in the legend (figure 1 - DO, GP).

Minor concerns:

- We miss some details on immunosuppressive drugs administered.

- References 3 and 13 are the same.

Comments on the Quality of English Language

I recommend manuscript revision by an English native speaker.

Author Response

Reviewer 1

In this study, the authors analyse a large cohort of patients with idiopathic pulmonary fibrosis (IPF), interstitial pneumonia with autoimmune features (IPF) and a peculiar IPAF subtype, namely UIPAF (IPAF with a UIP pattern on HRCT).

As compared to IPF, UIPAF and IPF displayed similar rate of antibodies and a similar rate of evolution towards a specific autoimmune disease (20-25%).

On the other hand, IPF and UIPAF were characterized by the use of antifibrotics and a propensity to progress according to the recent ATS-ERS criteria.

General comment:

Although the findings are incremental (UIP pattern is associated with worse outcome), this study explores an interesting and yet poorly explored area (UIPAF). I think the authors should address the following concerns:

Main concerns:

- Study population: considering the 2015 position paper, UIPAF should be considered as a subset of IPAF rather than a specific population. This point should be highlighted by the authors. Furthermore, it is striking that 22% of UIPAF have a positive rheumatoid factor and 11,7% positive ACPA, altogether pointing towards early rheumatoid arthritis. This point should be discussed.

REPLY

First of all, we are grateful for your time and for the valuable comment you provided in order to improve our manuscript. In particular we are grateful for this comment. Currently, based on the 2015 position paper those patients we called “UIPAF” cannot be considered to be a subset of IPAF, as only 1 of the 3 domains are satisfied. As reported in the introduction (line 70-74) and in the conclusion (line 360-362), based on our results, we suggest excluding the morphological domain, considering UIPAF patients to be IPAF. This point is important, and we tried to explain better this concept reporting in line 362 “…removing the morphological domain, thus classifying UIPAF and IPAF together in the same definition”. In view of this, we agree with you that UIPAF patients should be considered a subset of IPAF. As you appropriately noted, UIPAF patients had a significantly greater proportion of RF and a greater (although not statistically significant) proportion of ACPA than those reported by patients with IPAF. These autoantibodies suggest a possible progression towards RA, which is actually discussed in line 294-300.

- Study design: in its present form, it is not very clear whether this is an observational propspective study of a retrospective study. The authors should clarify this and provide a study FlowCHART.

REPLY

Thank you for your suggestion. The study has a prospective design, and now it is clearly reported in line 81. A flow-chart has been added (figure 1 and its legend, lines 167-170)

- A strong effort should be done on the clarity of figures: for example, Figure 4 is also noted as fiigure 3, some abbreviations are not defined in the legend (figure 1 - DO, GP).

REPLY

Thank you for your suggestion. We modified figures and legends in order to improve clarity, and specified all the variables taken into consideration

Minor concerns:

- We miss some details on immunosuppressive drugs administered.

REPLY

Thank you, we added a table (table 1) reporting the drugs used for each group of patients

- References 3 and 13 are the same.

REPLY

Thank you, fixed

Reviewer 2 Report

Comments and Suggestions for Authors

Thank you for inviting my review of this interesting 24 months follow up study comparing prognosis in significant numbers of IPAF, UIP-AF and IPF patients. The authors describe that about 25% of both IPAF and UIPAF groups progressed to fulfil criteria for a specific autoimmune disease (SAD) but with marked differences in the type of SAD shown between these groups. This fitted with the known preponderance of UIP in RA and GPA, as opposed to NSIP in PSS and IIM. I have a few observations and suggestions:

1 The title refers to 'prognosis' but most of the results focus on the pattern and progression of systemic auto-immune disease and their relationship to the subtype of lung disease, rather than specifically to survival. I strongly suggest that the title reflects this by replacing 'prognosis' with 'pattern and progression'.

2 In line 61, the authors state that UIPAF is 'very uncommon' but then that they recruited 69 patients with this. Perhaps 'less common' would be a more accurate term?

3 In many ways, the gist of the paper is that ILD patients with some serological or clinical autoimmune features will evolve towards a specific SAD at a rate of 1% per month, and that the HRCT subtype of lung disease is an important determinant of which SAD they may develop. The terminology is in danger of becoming overly complicated with the introduction of an increasing number of overlapping abbreviations, and perhaps the message can be kept simple.

4 Do mechanics hands predict a favourable or unfavourable prognostic association? It is hard to tell from the text.

5 In terms of true pulmonary prognosis, the absence of detailed data on treatment of this group reduces its value. The authors discuss other studies on therapy from line 288 but then admit to excluding most data from their own study. However there are clearly huge differences in their therapeutic approach between IPAF and UIPAF patients, with the former receiving mainly  immunosuppression and the latter anti-fibrotic therapy. Yet there is much data to suggest that immunosuppressives can still improve outcome in UIP patients. Hence any comparison of mortality between these groups would requires detail of therapy and the rationale behind this.

6 Finally, the language leaves a little to be desired, sometimes confusing the reader as to what the message may be. Early examples can be found in lines 53, 57, 62 and 73 but continue throughout the paper.

Comments on the Quality of English Language

6 Finally, the language leaves a little to be desired, sometimes confusing the reader as to what the message may be. Early examples can be found in lines 53, 57, 62 and 73 but continue throughout the paper.

Author Response

Reviewer 2

Thank you for inviting my review of this interesting 24 months follow up study comparing prognosis in significant numbers of IPAF, UIP-AF and IPF patients. The authors describe that about 25% of both IPAF and UIPAF groups progressed to fulfil criteria for a specific autoimmune disease (SAD) but with marked differences in the type of SAD shown between these groups. This fitted with the known preponderance of UIP in RA and GPA, as opposed to NSIP in PSS and IIM. I have a few observations and suggestions:

1 The title refers to 'prognosis' but most of the results focus on the pattern and progression of systemic auto-immune disease and their relationship to the subtype of lung disease, rather than specifically to survival. I strongly suggest that the title reflects this by replacing 'prognosis' with 'pattern and progression'.

REPLY

Thanks for the suggestion, we have changed the title so that it better reflects the main topic of the study.

2 In line 61, the authors state that UIPAF is 'very uncommon' but then that they recruited 69 patients with this. Perhaps 'less common' would be a more accurate term?

REPLY

Thank you, fixed

3 In many ways, the gist of the paper is that ILD patients with some serological or clinical autoimmune features will evolve towards a specific SAD at a rate of 1% per month, and that the HRCT subtype of lung disease is an important determinant of which SAD they may develop. The terminology is in danger of becoming overly complicated with the introduction of an increasing number of overlapping abbreviations, and perhaps the message can be kept simple.

REPLY

We agree with you. In our opinion the main topic of the manuscript is that UIPAF patients are almost identical to IPAF in their clinical presentation, serological features, PFTs at baseline and even after 12m-follow-up, and in the proportion of patients with an occult autoimmune disease. Anyway, in real life UIPAF patients are managed as IPF and their prognosis is extremely similar to IPF. The manuscript makes the simple proposal that, by removing the morphological domain from IPAF criteria, UIPAF patients could be considered as IPAF and probably managed accordingly, with the possibility of improving their prognosis. The terminology may seem complicated due to the presence of several abbreviations; however it should be considered that we analyzed more than 20 IPAF criteria, evaluating progression towards more than 10 different conditions. We tried to keep the number of abbreviations as low as possible, but a further reduction could weigh down the manuscript, which is already quite long.

4 Do mechanics hands predict a favourable or unfavourable prognostic association? It is hard to tell from the text.

REPLY

Thank you for your suggestion, actually the text is not clear. Now we reported “Finally, we evaluated the association between IPAF criteria and the PF-phenotype. We noted only a possible protective effect of Mechanic’s Hands in UIPAF patients (p=0.02, X2 7.2).”

5 In terms of true pulmonary prognosis, the absence of detailed data on treatment of this group reduces its value. The authors discuss other studies on therapy from line 288 but then admit to excluding most data from their own study. However there are clearly huge differences in their therapeutic approach between IPAF and UIPAF patients, with the former receiving mainly  immunosuppression and the latter anti-fibrotic therapy. Yet there is much data to suggest that immunosuppressives can still improve outcome in UIP patients. Hence any comparison of mortality between these groups would requires detail of therapy and the rationale behind this.

REPLY

We totally agree with your opinion. We added a table (table 1) including all the treatments used in the management of the three cohorts. The issue is that UIPAF is a research classification, but UIP patients without an alternative diagnosis should be clinically considered to be IPF and therefore immunosuppressive treatment in general would even considered to be detrimental (according to the PANTHER study) and therefore is not indicated. One of the main objectives of our work, as we wrote in the discussion, is actually this. UIPAF patients has almost the same clinical presentation as IPAF, and among UIPAF are included a similar proportion of occult autoimmune diseases to IPAF. However, in clinical practice, UIPAF patients are managed as IPF, sharing a similar prognosis. The question we tried to open up with our study is if UIPAF patients, being similar to IPAF, could be managed as IPAF instead of IPF, trying to improve the prognosis of these patients.

6 Finally, the language leaves a little to be desired, sometimes confusing the reader as to what the message may be. Early examples can be found in lines 53, 57, 62 and 73 but continue throughout the paper.

REPLY

Thank you, this new version of the manuscript has been proof-read by a native English speaker.

Round 2

Reviewer 2 Report

Comments and Suggestions for Authors

The authors have addressed my comments to my satisfaction

Comments on the Quality of English Language

The authors have addressed my comments to my satisfaction

Author Response

Thank you for your support in improving our manuscript

Kind Regards